# Multi-step coordination of telomerase recruitment in fission yeast through two coupled telomere-telomerase interfaces

Xichan Hu[†], Jinqiang Liu[†], Hyun-IK Jun, Jin-Kwang Kim, Feng Qiao*

Department of Biological Chemistry, University of California, Irvine School of Medicine, Irvine, United States

**Abstract** Tightly controlled recruitment of telomerase, a low-abundance enzyme, to telomeres is essential for regulated telomere synthesis. Recent studies in human cells revealed that a patch of amino acids in the shelterin component TPP1, called the TEL-patch, is essential for recruiting telomerase to telomeres. However, how TEL-patch—telomerase interaction integrates into the overall orchestration of telomerase regulation at telomeres is unclear. In fission yeast, Tel1[ATM]/Rad3[ATR]-mediated phosphorylation of shelterin component Ccq1 during late S phase is involved in telomerase recruitment through promoting the binding of Ccq1 to a telomerase accessory protein Est1. Here, we identify the TEL-patch in Tpz1[TPP1], mutations of which lead to decreased telomeric association of telomerase, similar to the phosphorylation-defective Ccq1. Furthermore, we find that telomerase action at telomeres requires formation and resolution of an intermediate state, in which the cell cycle-dependent Ccq1-Est1 interaction is coupled to the TEL-patch—Trt1 interaction, to achieve temporally regulated telomerase elongation of telomeres.

*For correspondence: qiao@uci.edu

[†]These authors contributed equally to this work

Competing interests: The authors declare that no competing interests exist.

## Introduction

Telomeres, the physical ends of linear chromosomes, are essential for stable maintenance of chromosomes by facilitating chromosome end replication and preventing them from being degraded or fusing with each other (*Artandi and Cooper, 2009*; *Palm and de Lange, 2008*). In most eukaryotes, telomeres are comprised of short tandem DNA repeats. Rather than a blunt end, the telomere consists of a 3' single-stranded overhang called the G-tail that provides the substrate for telomerase to counteract iterative telomere shortening after each round of DNA replication. Telomerase is a ribonucleoprotein enzyme that extends telomeres utilizing an RNA component as template for its reverse transcriptase protein subunit (named TERT in vertebrates and Trt1 in fission yeast) (*Autexier and Lue, 2006*; *Collins, 2006*). This telomerase-dependent nucleotide addition is rigorously limited to late S/G2 phase. Dysregulation of telomerase and the subsequent perturbation of telomere length homeostasis lead to severe defects in cell proliferation. As a result, a constellation of genetic diseases caused by mutations in the telomere maintenance machinery (*Sarek et al., 2015*), referred to as 'telomeropathies' (*Holohan et al., 2014*) or 'telomere syndromes' (*Armanios and Blackburn, 2012*), have been continuously identified, and include dyskeratosis congenita (DC), aplastic anemia, and multiple types of cancers.

Studies in telomerase regulation and telomere maintenance unveiled a precisely orchestrated process of telomerase recruitment (*Hockemeyer and Collins, 2015*; *Nandakumar and Cech, 2013*; *Schmidt and Cech, 2015*). In budding yeast, the interaction between the telomere-binding protein Cdc13 and the telomerase holoenzyme component Est1 regulates telomerase activation and its recruitment to the telomere (*Chandra et al., 2001*; *Evans and Lundblad, 1999*; *Pennock et al., 2001*; *Qi and Zakian, 2000*). Likewise, telomere-associated proteins in humans and fission yeast,

**eLife digest** The genetic blueprints for animals, plants and fungi are mostly contained within long strands of DNA and packaged into more compact thread-like structures called chromosomes. As such, most cells need to duplicate their chromosomes before they divide so that the two new cells each get a complete set of genetic instructions.

The machinery that copies DNA is unable to make it to the very ends of the chromosomes. Instead, an enzyme called telomerase adds new DNA to the chromosome ends to prevent them becoming too short. Problems with this process can cause serious issues, such as cell death or cancer, and so the activity of telomerase is carefully controlled. Other proteins guide telomerase to the ends of the chromosome only after the rest of the DNA has been copied. However, scientists do not know exactly how cells correctly time the arrival of telomerase.

A group of proteins called shelterin protects the chromosome ends, and studies with human cells have shown that telomerase attaches to a specific patch on one of shelterin proteins, called the TEL-patch, to begin its work. Now, Hu, Liu et al. have identified a similar TEL-patch in a shelterin protein from a type of yeast called fission yeast; this patch is also needed to attach telomerase to the chromosome ends. Further experiments with this yeast then showed that telomerase only arrives at the ends of the chromosomes after two parallel interaction interfaces have formed. Importantly, one of these interactions only takes place after most of the chromosomes are have been copied. As such, this "two-pronged interaction" mechanism ensures that the telomerase enzyme arrives at the end of the chromosomes at the right time.

Other similarities between human and fission yeast chromosome ends make it plausible that a comparable process controls the timing of telomerase attachment in human cells. However, more studies will be needed to confirm if this is the case.

which form a highly conserved protein complex called shelterin (*de Lange, 2005*; *Miyoshi et al., 2008*), are essential to regulate telomere states, recruit telomerase, and govern its activity on telomeres. In fission yeast, the shelterin complex contains the double-stranded DNA (dsDNA) binding protein Taz1 (the homolog of TRF1/TRF2 in humans) (*Cooper et al., 1997*) and the single-stranded DNA (ssDNA) binding protein Pot1 (human POT1 ortholog) (*Baumann and Cech, 2001*), which are bridged via direct protein-protein interactions between Rap1 (*Chikashige and Hiraoka, 2001*; *Kanoh and Ishikawa, 2001*), Poz1, and Tpz1 (human RAP1, TIN2 and TPP1 orthologs, respectively) (*Miyoshi et al., 2008*). The human system shares a conserved shelterin arrangement by forming a similar 'shelterin bridge' architecture (*Bianchi and Shore, 2008*). Recent studies discovered a cluster of residues on human TPP1, collectively termed TEL-patch, which mediates TPP1-TERT interaction to recruit telomerase to telomeres (*Nandakumar et al., 2012*; *Sexton et al., 2012*; *Zhong et al., 2012*). TEL-patch mediated TPP1-TERT interaction also confers increased telomerase processivity in vitro (*Nandakumar et al., 2012*; *Schmidt et al., 2014*). Moreover, a detailed genetic study of TEL-patch mutants revealed multiple functions of TPP1 in telomerase recruitment, activation, and telomere length feedback regulation (*Sexton et al., 2014*). Altered TEL-patch function is clinically manifested as Hoyeraal-Hreidarsson (HH) syndrome. HH patients bear extremely short telomeres, even in comparison to other DC patients. A germline mutation causing a single-amino-acid deletion (K170Δ) in the human TPP1 TEL-patch that affects its telomerase recruitment function has been identified as the causal mutation of HH (*Kocak et al., 2014*). This signifies the critical telomerase recruitment function of the TEL-patch in normal stem cell development.

Interestingly, fission yeast *Schizosaccharomyces pombe*, which has a similar shelterin architecture to mammals, contains an additional shelterin component called Ccq1 (*Flory et al., 2004*), the functional homolog of which has not yet been identified in mammals. Ccq1 was demonstrated to be a cell cycle-dependent telomerase recruitment factor (*Miyoshi et al., 2008*; *Tomita and Cooper, 2008*). Further investigation showed that Ccq1 is required to bring telomerase to telomeres through its Tel1$^{ATM}$/Rad3$^{ATR}$-mediated Thr93 phosphorylation during late S phase, which creates a binding site for the 14-3-3 domain of Est1 (*Moser et al., 2011*; *Webb and Zakian, 2012*; *Yamazaki et al., 2012*). Est1 is an accessory protein of telomerase holoenzyme and is linked to

the telomerase protein subunit (Trt1) via their co-association with telomerase RNA (TER1) (*Leonardi et al., 2008*). Ccq1 also interacts with Tpz1 (*Miyoshi et al., 2008*), and the Ccq1-Tpz1 interaction is therefore thought to bring the telomerase complex (Est1-TER1-Trt1) to the telomere (*Moser et al., 2009*) via the Ccq1 Thr93-phosphorylation dependent Ccq1-Est1 interaction. However, in-depth analyses further revealed that Ccq1-Est1 and Ccq1-Tpz1 interactions seem to be mutually exclusive (*Armstrong et al., 2014*). Moreover, Est1-Ccq1 interaction could be disrupted by TER1 in a yeast three-hybrid analysis and Est1 mutations that affect Est1-TER1 interaction also impair Est1-Ccq1 interaction (*Armstrong et al., 2014*; *Webb and Zakian, 2012*). In addition, based on a Ccq1-centric model, it is hard to explain why the telomeric association of Est1 requires not only Ccq1, but also Trt1 and TER1, which are downstream from Ccq1-Est1 interaction (*Webb and Zakian, 2012*). Therefore, the hypothetical Tpz1-Ccq1-Est1-TER1-Trt1 interaction chain seems unlikely to form to mediate telomerase recruitment. These results imply that an alternative mechanism exists to directly associate telomerase to other shelterin components, such as Tpz1, to initiate telomere elongation in fission yeast in response to the cell cycle-dependent Ccq1-Thr93 phosphorylation (*Chang et al., 2013*).

In this study, we identified a Tpz1 mutation in the TEL-patch region that results in an ever shorter telomere (*EST*) phenotype, similar to the telomere phenotype of human TPP1 TEL-patch mutants. We observed decreased telomeric association of Trt1 and weakened Tpz1-Trt1 interaction in this Tpz1 TEL-patch mutant, indicating the conserved role of the TEL-patch in telomerase recruitment. Our epistasis analyses demonstrated that the Tpz1 TEL-patch functions by positively regulating Trt1. Furthermore, we found that telomerase action at telomeres requires formation and resolution of an intermediate state, formed via two cooperative telomere-telomerase interfaces involving cell cycle-regulated Ccq1-Est1 interaction and Tpz1 (TEL-patch)-Trt1 interaction. As a result, the temporal information for telomerase recruitment is endowed to the TEL-patch through the phosphorylation status of Ccq1 Thr93, thus achieving cell cycle-specific telomere elongation.

## Results

### Tpz1 TEL-patch mutations lead to *Ever Shorter Telomere—EST* phenotype

Telomerase exists in low abundance in the cell. Therefore, the interaction between shelterin and telomerase has been proposed to enrich telomerase at chromosome ends. Indeed, a group of surface-exposed amino acids in the human TPP1 OB-domain, termed TEL-patch, are found to be necessary for the telomerase recruitment to telomeres (*Nandakumar et al., 2012*; *Sexton et al., 2012*; *Zhong et al., 2012*). Although the fission yeast shelterin component Ccq1 has been connected to telomerase recruitment in this model organism, it is unlikely to be the sole factor to link Trt1 to telomeres (*Armstrong et al., 2014*). Additional interactions between shelterin components and Trt1 must exist to bring Trt1 directly to the telomere. Given the high conservation of the OB-fold domain arrangement in the N-terminal regions of *S. pombe* Tpz1 and human TPP1, we decided to test whether a Tpz1 TEL-patch functions in *S. pombe* as an interface between telomerase and shelterin. To identify candidate residues for such a TEL-patch in *S. pombe* Tpz1, we performed a sequence alignment between fission yeast Tpz1 and human TPP1 in combination with a secondary structure prediction (*Figure 1A*). We then identified 12 conserved Tpz1 residues in the region corresponding to the human TPP1 TEL-patch as candidate residues for fission yeast Tpz1 TEL-patch. *S. pombe* cells bearing individual mutations in these 12 residues of Tpz1 were tested for their telomere maintenance. *tpz1-I105R* and *tpz1-V107R* mutant strains appear to have destabilized Tpz1 protein and display a similar telomere deprotection phenotype (*Figure 1—figure supplement 1*) to *tpz1Δ* strain. The remaining 10 *tpz1* mutant strains have comparable levels of Tpz1 expression as the wild-type strain (*Figure 1—figure supplement 2*). Strains bearing *tpz1-T73A*, *tpz1-K75E*, *tpz1-R76E*, *tpz1-I77R*, and *tpz1-R81E* displayed shortening telomeres (*Figure 1B* and *Figure 1—figure supplement 1*), implicating positive regulatory roles of these residues in telomere length regulation. Although *tpz1-T73A*, *tpz1-K75E*, and *tpz1-R76E* cells all have shortened telomeres, in contrast to the *tpz1-I77R and tpz1-R81E* cells, their telomeres remain stably short (*Figure 1B* and *Figure 1—figure supplement 1*). This telomere phenotype is similar to that of previously identified fission yeast *tpz1-K75A* (*Armstrong et al., 2014*) or human TPP1-L104A

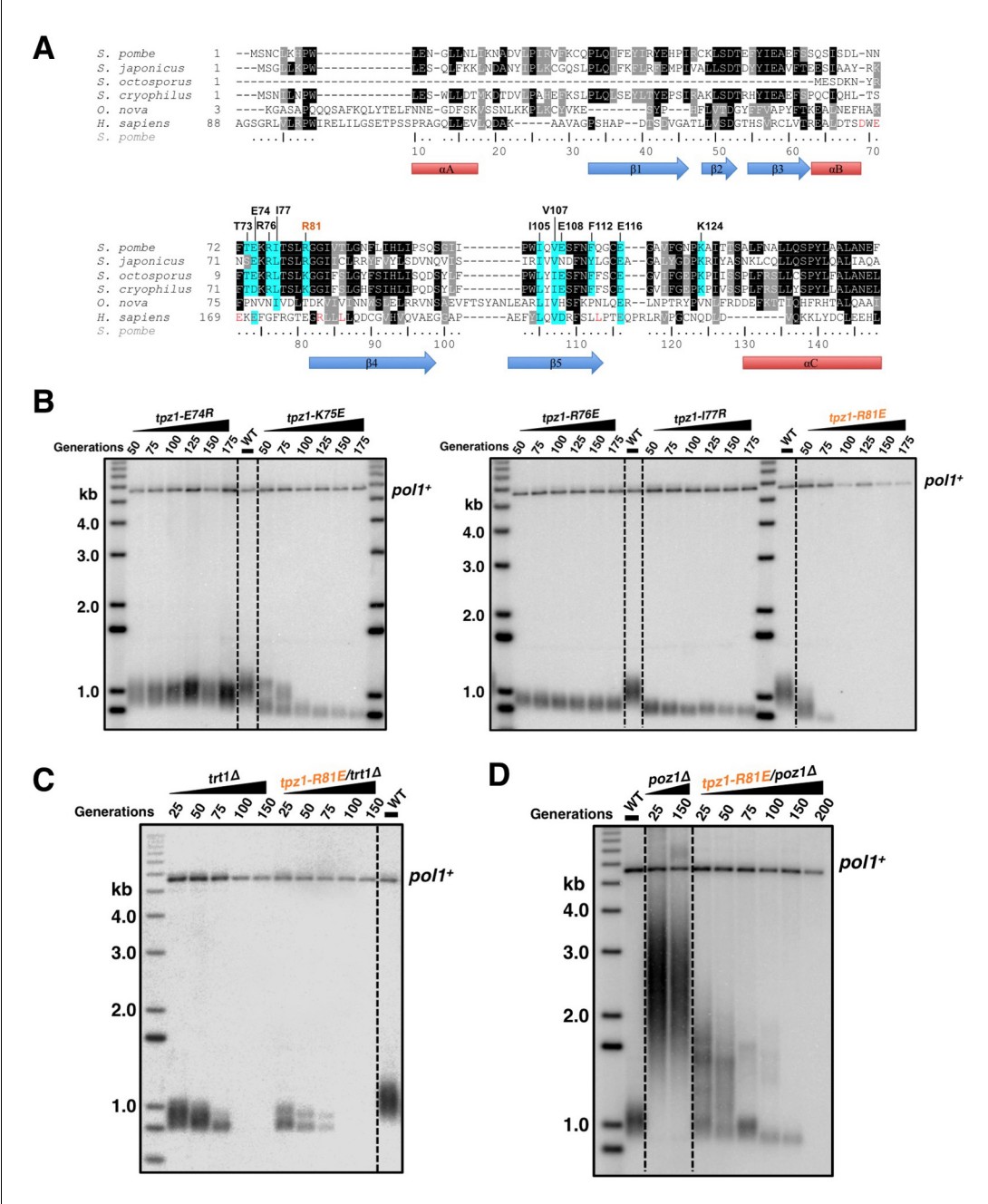

**Figure 1.** Fission yeast Tpz1 TEL-patch mutant leads to an EST phenotype. (**A**) A sequence alignment of OB fold-domains between fission yeast Tpz1, human TPP1 and *Oxytricha nova* TEBP-β, in combination with a secondary structure prediction. Twelve conserved fission yeast Tpz1 residues subjected to mutagenesis are highlighted in blue with their identities and sequence numbers indicated above the alignment. Residues colored in red are previously identified TEL-patch residues in human TPP1 and mutations of them lead to compromised TPP1-TERT interaction. Black and grey shading indicates sequence identity and similarity, respectively. If the sequence is identical among at least 50% of species, the residues will be shaded in black. The same rule applies to sequence similarity, which is shaded in grey. (**B**) Southern blot analysis to measure telomere lengths using EcoRI-digested genomic DNA visualized by the telomere DNA probe for the indicated *tpz1* mutant strains from successive re-streaks on agar plates. *tpz1-K75E, tpz1-R76E, tpz1-I77R, and tpz1-R81E* strains caused telomere shortening and *tpz1-R81E* cells showed the classic *Ever Shorter Telomere* (*EST*) phenotype. For the telomere length analysis southern blots presented in all the figures, the 1 kb plus marker from Invitrogen is used. Wild type cells are denoted as 'WT' in the blot. *pol1+* indicates the EcoRI-digested *pol1+* DNA fragment used as a loading control. (**C**) Double-mutant strain *tpz1-R81E/trt1Δ* shows *EST* telomere phenotype, similar to the *trt1Δ* single-mutant strain. (**D**) Double-mutant strain *tpz1-R81E/poz1Δ* shows progressive telomere shortening phenotype.

*Figure 1 continued on next page*

*Figure 1 continued*

The following figure supplements are available for figure 1:

**Figure supplement 1.** Telomere length measurement of *tpz1* mutant strains.

**Figure supplement 2.** Evaluation of Tpz1 expression levels in *tpz1* mutant strains.

**Figure supplement 3.** The later generation *tpz1-R81E/poz1Δ* mutant strain forms circularized chromosomes.

(*Sexton et al., 2014*), proposed to affect telomerase activation and telomere length homeostasis set point, respectively. Strikingly, *tpz1-R81E* cells showed the classic <u>E</u>ver <u>S</u>horter <u>T</u>elomere (**EST**) phenotype (*Lundblad and Szostak, 1989*), similar to *trt1Δ* cells (*Figure 1B*) (*Nakamura et al., 1997*). The deterioration in telomere maintenance of *tpz1-R81E* strain started as early as ~50 generations. Its telomeres shorten to the critical length at 75 generations, and are almost completely lost afterwards. Gradual telomere loss was also observed in *tpz1-I77R* cells, albeit to a much milder degree than in *tpz1-R81E* cells. Furthermore, Tpz1-Arg81 and Trt1 appear to have an epistatic relationship because telomere shortening in the *tpz1-R81E/trt1Δ* double mutant is not additive (*Figure 1C*). The similar phenotypes observed for *tpz1-R81E*, *trt1Δ*, and *tpz1-R81E/trt1Δ* support a role of Tpz1-Arg81 in directly regulating telomerase activity on telomeres. Similar to the previously identified *tpz1-K75A* mutant that is defective in telomerase activation (*Armstrong et al., 2014*), *tpz1-K75E* and *tpz1-R76E* mutants have a milder telomere loss phenotype and their shortened telomeres are maintained for many generations. These observations clearly distinguish Tpz1-Arg 81 from telomerase activation residues in Tpz1, i.e. Lys75 or Arg76.

We next asked whether the requirement for TEL-patch residue Tpz1-Arg81 can be bypassed by eliminating the negative shelterin linkage, which keeps telomeres constitutively in the telomerase-extendible state (*Jun et al., 2013*). To this end, we constructed a double-mutant strain, *tpz1-R81E/poz1Δ*, in which *poz1Δ* leads to defective shelterin linkage. As shown in *Figure 1D*, *tpz1-R81E/poz1Δ* cells presented *EST* phenotype and the cells senesced at a later generation, similar to the *trt1Δ/poz1Δ* mutant (*Miyoshi et al., 2008*). A subpopulation survived at ~200 generations by circularizing their chromosomes to bypass the need for telomerase (*Figure 1D* and *Figure 1—figure supplement 3*). This result clearly indicates that Tpz1-Arg81 functions downstream of telomere switching from the non-extendible to the extendible state and upstream of telomerase action, most likely to mediate telomere-telomerase interaction. Therefore, we genetically demonstrated the existence of the TEL-patch in fission yeast Tpz1 that is functionally analogous to the TEL-patch of human TPP1.

## The Tpz1 TEL-patch contributes to Tpz1-Trt1 interaction

Mutations of the TEL-patch residues in human TPP1 disrupt the direct interaction between TPP1 and TERT. Moreover, the TEN domain of human TERT was demonstrated to mediate its interaction with TPP1 (*Schmidt et al., 2014*), providing an interface required for the recruitment of telomerase to telomeres. In fission yeast, both Tpz1 and Ccq1 have been found to interact with Trt1 independent of telomerase RNA (*Armstrong et al., 2014*; *Tomita and Cooper, 2008*). To directly assess the function of the putative TEL-patch residue Tpz1-Arg81, we first examined the binding efficiency between Tpz1-R81E and Trt1 utilizing co-immunoprecipitation assay. As shown in *Figure 2A,B* and *Figure 2—figure supplement 1*, whereas Tpz1-E74R, Tpz1-K75A, Tpz1-K75E, and Tpz1-R76E pulled down similar amounts of Trt1 as the wild-type Tpz1, Tpz1-R81E only immunoprecipitated 30% of Trt1 compared to the wild-type Tpz1 (*Figure 2B and C*). Considerably reduced Tpz1-Trt1 interaction in the *tpz1-R81E* mutant correlates well with its progressive telomere shortening phenotype (*Figure 1B*), further supporting an essential role that Tpz1-Arg81 plays in mediating the interaction between telomerase and shelterin at the telomere. Interestingly, *tpz1-I77R* cells lost about 50% of the Tpz1-Trt1 interaction (*Figure 2B and C*). This observation suggests that Tpz1-Ile77 is likely to be part of the TEL-patch as well, consistent with the compromised telomere maintenance in *tpz1-I77R* cells, albeit milder than that of the *tpz1-R81E* cells (*Figure 1B*). The *tpz1-R76E* mutant retains the wild-type Tpz1-Trt1 interaction (*Figure 2A*) but displays stably shortened telomeres. This implies

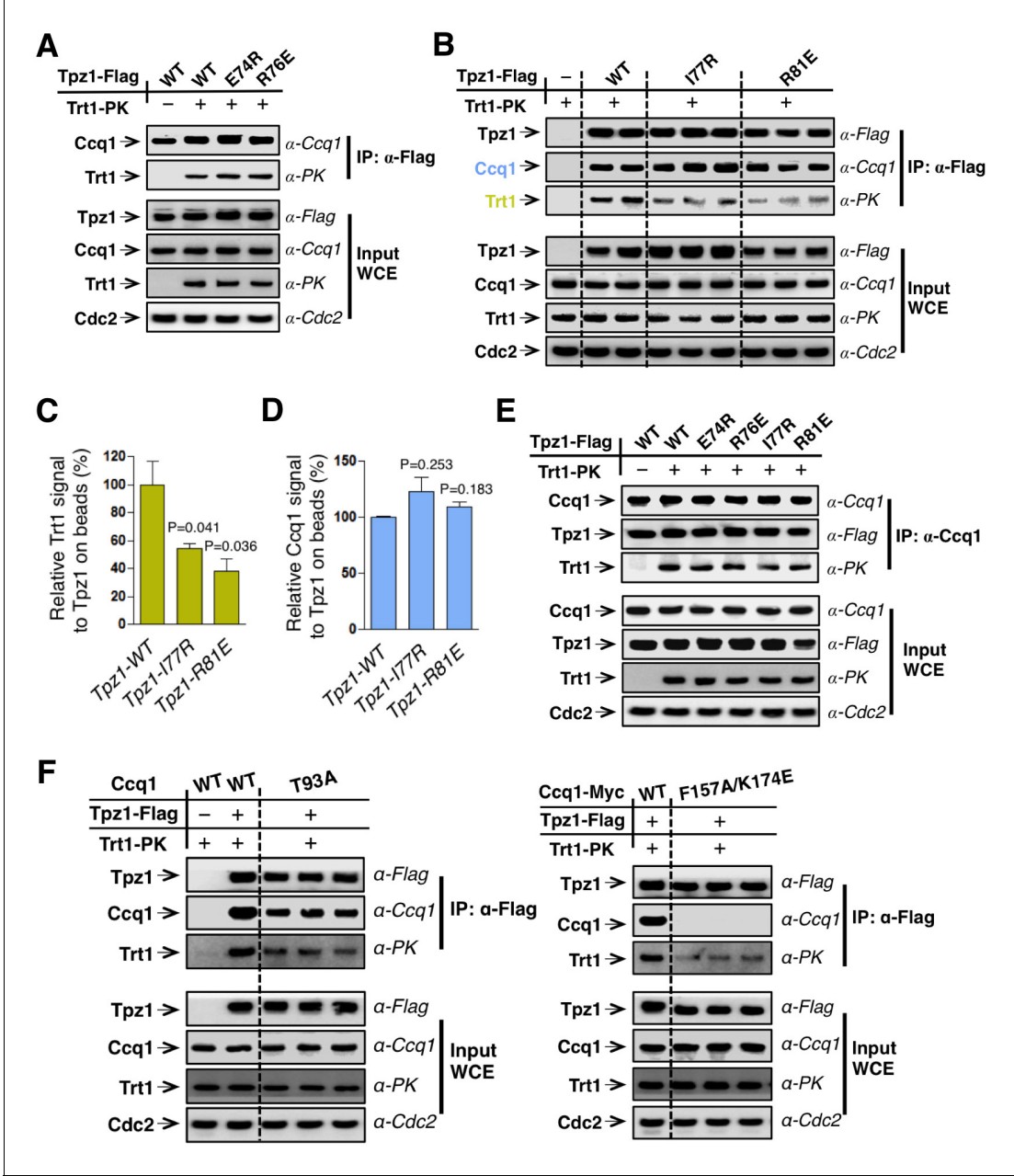

**Figure 2.** The Tpz1 TEL-patch mutant is defective in Tpz1-Trt1 interaction. (A) Co-immunoprecepitation (Co-IP) assays evaluating bindings between Trt1 and Tpz1-E74R and Tpz1-R76E. Cdc2 was shown as the loading control. Input: 1/30 of input WCE (whole cell extract). (B) Co-IP assays showing that Tpz1-I77R and Tpz1-R81E significantly decrease interaction between Tpz1 and Trt1. Cdc2 was shown as the loading control. Input: 1/30 of input WCE. (C) Quantification of the binding efficiency between Trt1 and Tpz1 mutants from (B). The interaction between Trt1 and Tpz1-WT is set to be 100%. Trt1 levels in the IP were normalized to Tpz1 bound to the beads. (D) Quantification of the binding efficiency between Ccq1 and Tpz1 mutants from (B). The interaction between Ccq1 and Tpz1-WT is set to be 100%. Ccq1 levels in the IP were normalized to Tpz1 bound to the beads. (E) Co-IP assays evaluating the binding efficiency between Trt1 and Ccq1 in various Tpz1 mutant cells. Cdc2 was shown as the loading control. Input: 1/30 of input WCE. (F) Co-IP assays evaluating the binding efficiency between Trt1 and Tpz1 in *ccq1-T93A* and *ccq1-F157A*/K174E. Cdc2 was shown as the loading control. Input: 1/30 of input WCE.

The following figure supplement is available for figure 2:

**Figure supplement 1.** Trt1-Tpz1 interaction is intact in *tpz1-K75E* and *tpz1-K75A* strains.

that Tpz1-Arg76 is involved in telomerase activation, similar to what was previously described for Lys75 (*Figure 2—figure supplement 1*) (*Armstrong et al., 2014*).

If the Tpz1-Trt1 interaction were mediated solely via Ccq1, the reduced Tpz1-Trt1 interaction observed in *tpz1-R81E* cells would most likely result from reduced Tpz1-Ccq1 interaction, because other components in the proposed Tpz1-Ccq1-Est1-Trt1 interaction-chain remain intact. However, as shown in *Figure 2B,D and E*, all tested Tpz1 mutants, including Tpz1-R81E, interact with Ccq1 similarly to wild-type Tpz1. Moreover, the Ccq1-Trt1 interaction remains unchanged for all Tpz1 mutants as well (*Figure 2E*). Next, we tested whether two Ccq1 mutants, Ccq1-F157A/K174E and Ccq1-T93A, previously shown to be defective in Tpz1-Ccq1 (*Liu et al., 2015*) and Ccq1-Est1 (*Moser et al., 2011*) interactions, respectively, can completely abolish Tpz1-Trt1 interaction. As shown in *Figure 2F*, Tpz1-Trt1 interaction is clearly retained in both Ccq1-F157A/K174E and Ccq1-T93A mutants. This was particularly striking for the Ccq1-F157A/K174E mutant, in which Tpz1-Ccq1 interaction is completely disrupted. These data, together with previous studies (*Armstrong et al., 2014*; *Webb and Zakian, 2012*), argue against Ccq1 as the sole platform that recruits telomerase to telomeres. These results further point to the existence of the TEL-patch in fission yeast Tpz1 and indicate a conserved function in the recruitment of telomerase.

## The TEL-patch mutant of Tpz1 fails to localize telomerase to telomeres

To further test whether the failure of the *tpz1-R81E* mutant to replenish telomeres emanates from its inability to recruit telomerase to telomeres, we directly tested the group of Tpz1 OB-fold domain mutants for in vivo localization of telomerase to telomeres using ChIP assays. As expected, *tpz1-R81E*, which has significantly reduced Tpz1-Trt1 interaction, exhibited a dramatic decrease in the association of Trt1 with telomeres (*Figure 3A*). In contrast to the *tpz1-R81E* mutant, a telomerase activation-defective mutant *tpz1-R76E* displayed wild-type telomerase localization to telomeres (*Figure 3A*). The latter result is consistent with the phenotype of another previously described telomerase activation mutant strain *tpz1-K75A* (*Armstrong et al., 2014*). Interestingly, the *tpz1-I77R* mutant strain also showed significant decreased localization of Trt1 to telomeres (*Figure 3A*), consistent with the decreased Tpz1-Trt1 interaction observed in the co-immunoprecipitation assay (*Figure 2B*). In contrast, in all the tested Tpz1 OB-fold domain mutants, little effect was observed on the telomeric association of both Tpz1 and Ccq1 (*Figure 3B and C*). These results directly reveal a specific role for Tpz1 N-terminal OB-domain residues Ile77 and Arg81 in mediating telomere-Trt1 interaction and Trt1 recruitment. Moreover, since *tpz1-I77R* cells have a much more subtle telomere shortening phenotype than *tpz1-R81E* cells, we speculate that a threshold amount of residual telomerase interaction may be required for telomere maintenance upon disruption of the Tpz1 TEL-patch. Taken together, we redefined the TEL-patch in fission yeast Tpz1, which functions analogously to the previously characterized TEL-patch in human TPP1 (*Nandakumar et al., 2012*; *Sexton et al., 2012*; *Zhong et al., 2012*).

## Fusing Trt1 to Tpz1 bypasses the requirement for functional Tpz1 TEL-patch

Direct fusion of telomerase to a shelterin component has been shown to rescue the telomerase recruitment defect, but not other defects, such as telomere activation or telomere length homeostasis regulation in both fission yeast (*Armstrong et al., 2014*) and human embryonic stem cells (hESC) (*Sexton et al., 2014*). We next tested whether the inability of the Tpz1 TEL-patch mutant to maintain telomere length could be rescued by forcing telomerase to physically associate with telomeres. To this end, we measured telomere lengths of strains with Trt1 fused to Tpz1 mutants as previously described (*Armstrong et al., 2014*). Apparently, the strain bearing fused Trt1—Tpz1 TEL-patch mutant (*trt1—tpz1-R81E*) maintained the same telomere length as the *trt1—tpz1* wild-type strain (*Figure 4A*), indicating rescue of telomere shortening. However, fusion of Trt1 with Tpz1-L449A, which disrupts the Tpz1-Ccq1 interaction, failed to restore telomere maintenance, as previously reported (*Armstrong et al., 2014*). The expression levels of all Trt1—Tpz1 fusion proteins appear to be similar (*Figure 4B*). These results suggest that the Tpz1-Ccq1 interaction is required for aspects of telomere elongation besides bridging telomerase to telomeres. This interpretation is consistent with our previous genetic study implicating Ccq1 in switching telomeres from non-extendible to extendible state (*Jun et al., 2013*).

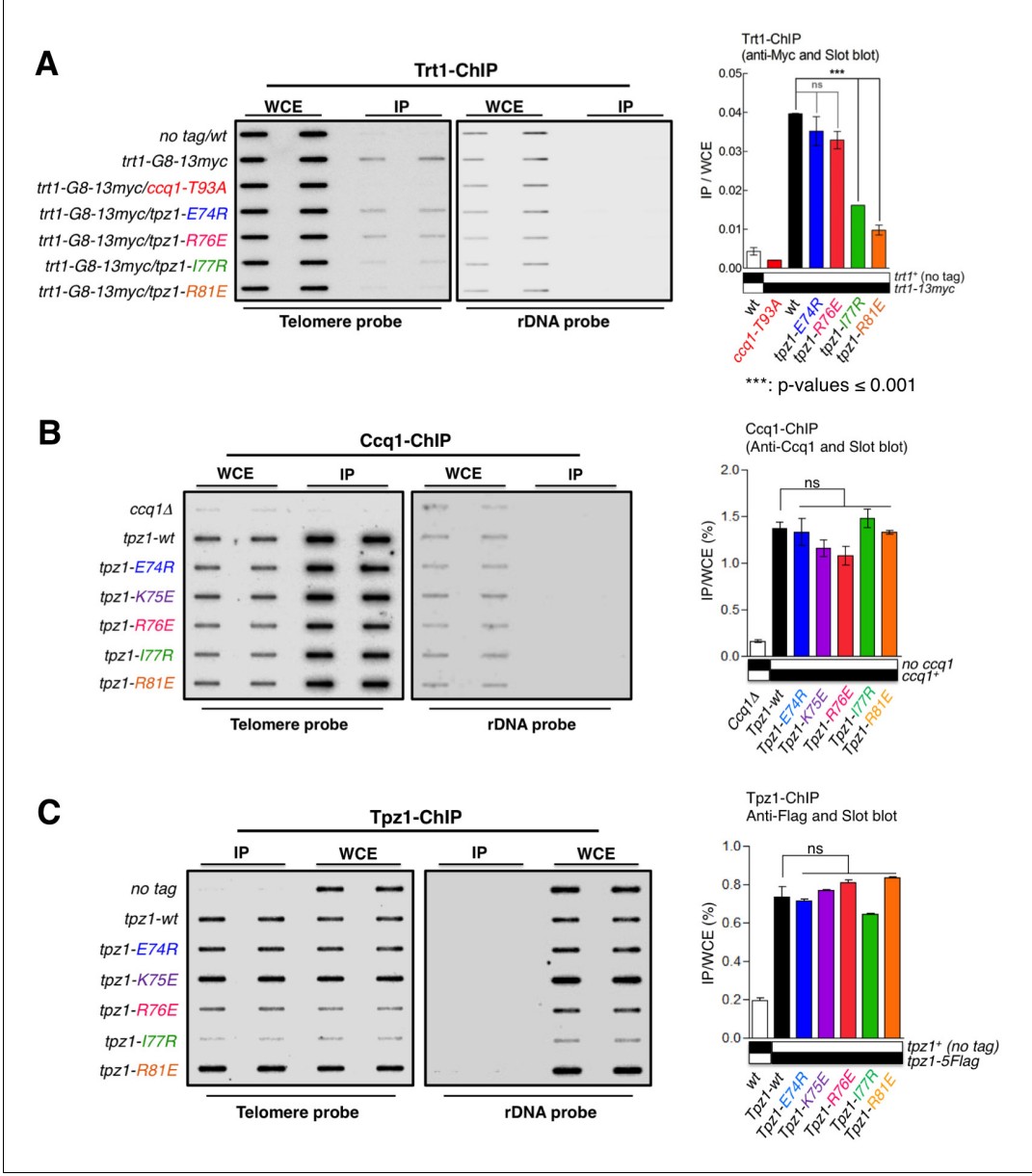

**Figure 3.** The Tpz1 TEL-patch mutant fails to localize telomerase to telomeres. (A–C) Enrichment of Trt1 (A), Ccq1 (B) or Tpz1 (C) at telomeres is monitored by chromatin immunoprecipitation (ChIP) assay. Slot-blot was used to visualize telomere association of Trt1, Ccq1 or Tpz1 in each indicated *tpz1* mutant strains. Error bars in the quantitation of the slot-blot analysis represent standard deviations of two individual repeats.

## Est1 and Ccq1 form a stable complex through two binding sites

In fission yeast, Rad3[ATR]/Tel1[ATM]-dependent phosphorylation of Ccq1 at Thr93 promotes direct interaction between Ccq1 and Est1—a subunit of the telomerase holoenzyme (*Moser et al., 2011*; *Yamazaki et al., 2012*). The 14-3-3–like domain of Est1 was shown to recognize Ccq1 phosphorylated at Thr93 and was proposed to enable telomerase-telomeres association via the Ccq1-Tpz1 complex (*Moser et al., 2011*). Different from the proposed Tpz1-Ccq1-Est1 chain-interaction that connects telomerase to telomere shelterin, Ccq1-Est1 and Ccq1-Tpz1 interaction were demonstrated to be mutually exclusive (*Armstrong et al., 2014*) and the binding surfaces for Est1 and Tpz1 in Ccq1 are most likely to overlap. Our recent work utilizing a new strategy, called *MICro-MS* (Mapping Interfaces via Crosslinking-Mass Spectrometry), mapped the Tpz1-interacting interface on

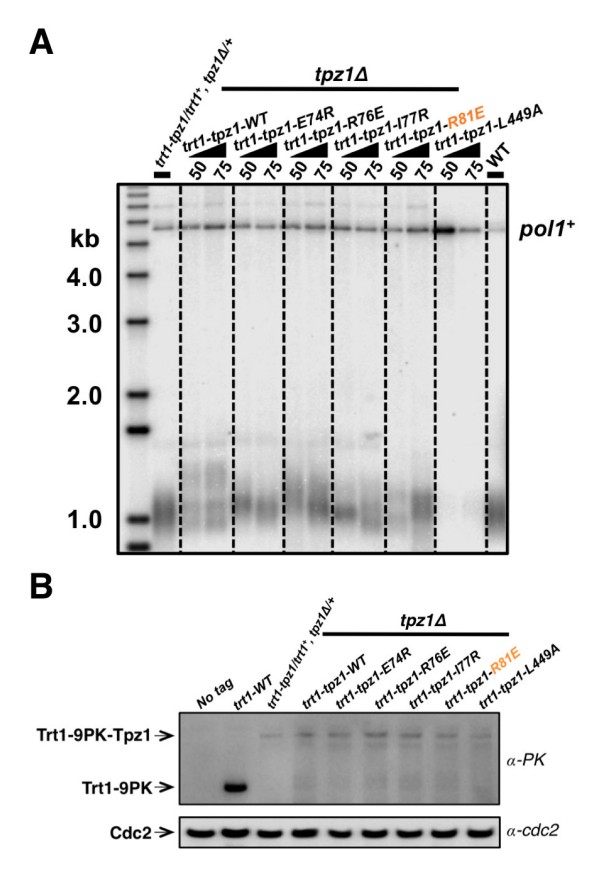

**Figure 4.** Fusing Trt1 to Tpz1 bypasses the requirement for functional Tpz1 TEL-patch. (**A**) Southern blot analysis to measure telomere lengths using EcoRI-digested genomic DNA visualized by the telomere DNA probe for the *trt1-tpz1* fusion strains with indicated WT or mutant versions of *tpz1*. 9PK tags were inserted between *trt1* and *tpz1*. (**B**) Western blot showing expression levels of PK-tagged Trt1 protein and the chimeric Trt1-Tpz1 fusion proteins. Cdc2 was used as a loading control for total proteins.

Ccq1 and isolated a group of separation-of-function mutants that specifically disrupt Ccq1-Tpz1 interaction (*Liu et al., 2015*). If Ccq1 interacts with Est1 via an overlapping surface that also mediates Ccq1-Tpz1 interaction, Ccq1 mutants that are defective in Ccq1-Tpz1 interaction is most likely to compromise Ccq1-Est1 interaction too. Indeed, as shown in *Figure 5A*, Ccq1-F157A/K174E and Ccq1-I175R, which were identified previously to be defective in Tpz1-Ccq1 interaction, both significantly diminished Ccq1-Est1 interaction in co-immunoprecipitation assays, whereas Ccq1-V152R and Ccq1-L177R still retained wild-type binding to Est1. Unexpectedly, we found that Ccq1-T93A, the Ccq1 mutant in the Rad3[ATR]/Tel1[ATM] phosphorylation site Thr93, did not completely abolish Ccq1-Est1 interaction; rather, it diminished Ccq1-Est1 interaction to a similar degree that was observed for Ccq1-F157A/K174E or Ccq1-I175R mutant.

We therefore speculated that Ccq1 and Est1 probably interact with each other via two binding sites: one provided by phosphorylated Thr93 and the other by the HDAC2/3-like domain at the N-terminus of Ccq1 (as depicted in *Figure 5B*). As a result, mutating either site only partially disrupts Ccq1-Est1 interaction. Indeed, as expected from the two-binding-site model, a Ccq1 mutant with both binding sites mutated—Ccq1-T93A/F157A/K174E completely lost its ability to bind to Est1 (*Figure 5C*). Previous mutational analysis of Est1 using the yeast two-hybrid assay identified residues in the predicted Est1 phospho-binding site (R79 and R180) that mediate the interaction between Est1 and Thr93-phosphorylated Ccq1 (*Moser et al., 2011*). Another study found that an additional residue—K252 in the 14-3-3–like domain of Est1 is also important for Est1-Ccq1

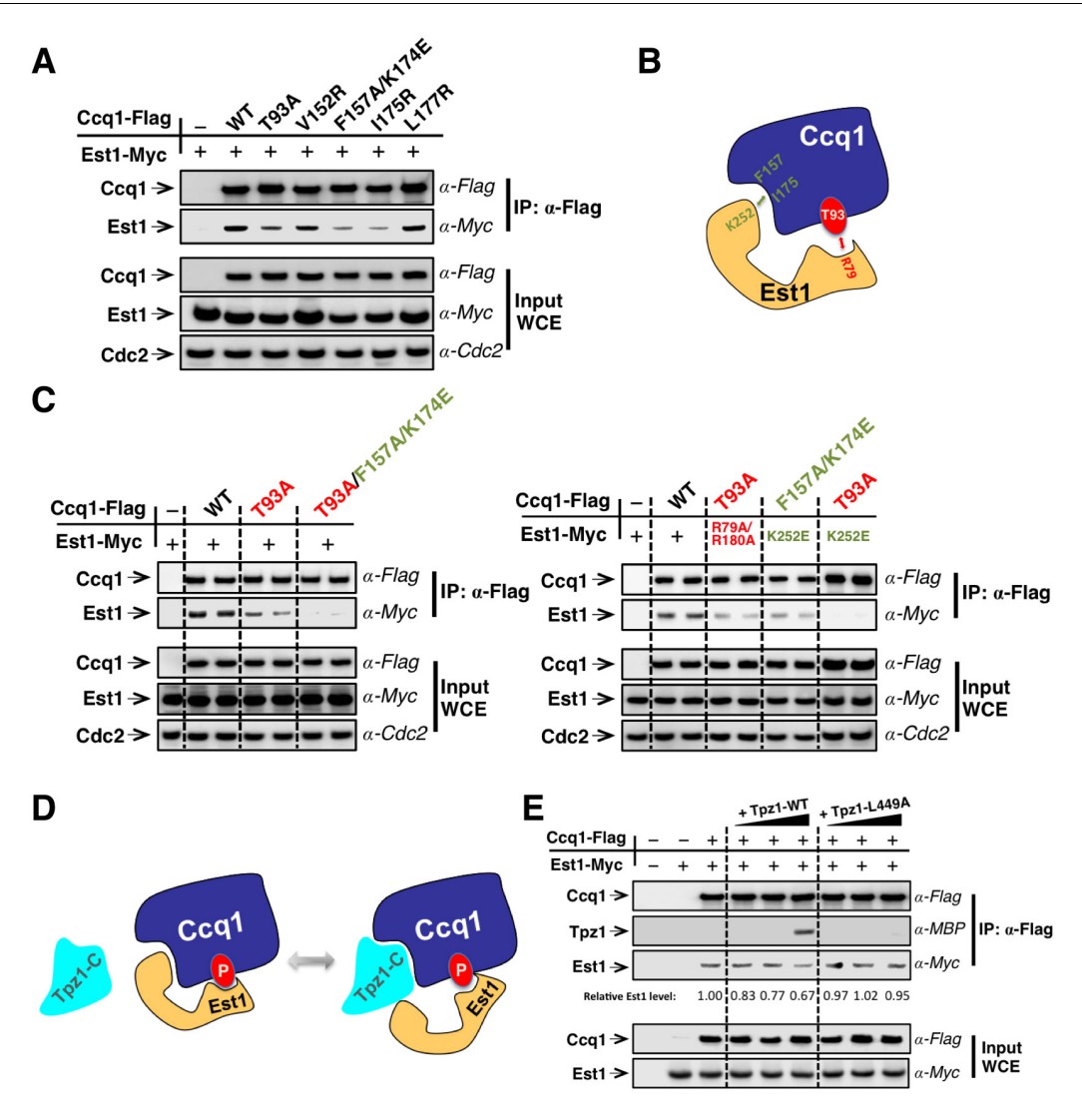

**Figure 5.** Est1 binds to Ccq1 through two different binding sites. (A) Ccq1-Est1 interaction is diminished but not completely disrupted in the *ccq1-T93A, ccq1-F157A/K174E*, and *ccq1-I175R* strains as evaluated by Co-IP assays. Cdc2 was shown as the loading control. Input: 1/30 of input WCE. (B) Schematic model of Est1 and Ccq1 interaction showing two binding sites for their interaction as further tested in (C). (C) Co-IP assays evaluating the contribution of two binding sites to the binding between Est1 and Ccq1. Cdc2 was shown as the loading control. Input: 1/30 of input WCE. (D) Schematic model of Tpz1 C-terminal domain, Est1 and Ccq1 forming an intermediate Est1-Ccq1-Tpz1 complex. (E) Competitive binding assay showing that Tpz1 can partially compete with Est1 for its interaction with Ccq1; however, the Tpz1-Ccq1 interaction defective mutant Tpz1-L449A cannot.

interaction, besides for Est1-TER1 interaction (*Webb and Zakian, 2012*). In the Est1 structure model, K252 is on a different surface compared to phospho-Thr93 binding residue R79 and R180. Est1-K252 is therefore very likely to be part of the binding site for the HDAC2/3-like domain at the N-terminus of Ccq1. In fact, in *ccq1-T93A/est1-K252E* cells, which bear mutations on Ccq1 and Est1 that in combination disrupt both binding sites, Ccq1-Est1 interaction could not be detected (*Figure 5C*). However, for strains *ccq1-T93A/est1-R79A/R180A* and *ccq1-F157A/K174E/est1-K252E*, in which mutations were introduced to only one of the Ccq1-Est1 binding sites with the other site intact, Ccq1 and Est1 interaction could still be observed at similar levels to single mutants *ccq1-T93A* or *est1-K252E* (*Figure 5C*). Altogether, our results indicate that phospho-Thr93 in Ccq1

contributes to one of the two binding sites that mediate Ccq1-Est1 interaction. The second Est1-interacting surface resides in the HDAC2/3-like domain at the N-terminus of Ccq1, and overlaps with the Tpz1-interacting surface.

To further understand the relationship of Tpz1 and Est1 binding to the HDAC2/3-like domain in Ccq1, we performed a competitive binding assay. This evaluated the interaction between Est1 and Thr93-phosphorylated Ccq1 in the presence of Tpz1. In this assay, Flag-tagged Ccq1 was immuno-precipitated from the whole cell extract by α-Flag agarose beads. We then titrated increasing amounts of MBP-Tpz1-CTD to the Ccq1-Est1 complex. As shown in *Figure 5E* the addition of MBP-Tpz1-CTD partially displaced the Ccq1-bound Est1 at a high concentration of Tpz1. This did not occur when the Ccq1-binding defective mutant MBP-Tpz1-CTD L449A was added. Because of the entropic advantage provided by the phospho-Thr93-centered Est1-Ccq1 binding site, a high concentration of Tpz1 is needed to compete Est1 off the HDAC2/3-like domain of Ccq1. Taken together, the mutational analyses and the competitive binding assays indicate that Est1-Ccq1 inter-action and Tpz1-Ccq1 interaction are mutually exclusive on the HDAC2/3-like domain of Ccq1 since both association events utilize the same binding surface on Ccq1 containing Phe157 and Lys174, et al. In addition, because phosphorylated Ccq1-Thr93 provides a secondary binding site for Ccq1-Est1 association, Tpz1-Ccq1-Est1 ternary complex can also form, likely in a transient man-ner, not as stable as Tpz1-Ccq1 or Ccq1-Est1 binary complexes. Thus, combined with our discov-ery of the TEL-patch in the N-terminal OB-domain of Tpz1, we suspect that the cell cycle-regulated Ccq1 (phospho-Thr93)-Est1 interaction can be coupled to the Tpz1 (TEL-patch)-Trt1 interaction via the Tpz1-Ccq1-Est1 intermediate complex to coordinate the recruitment of telome-rase to telomeres.

## TEL-patch function is regulated in a cell cycle-dependent manner

As shown before, Rad3[ATR]/Tel1[ATM]-dependent phosphorylation of Ccq1 at Thr93 is cell cycle-regu-lated and peaks during late S phase (*Chang et al., 2013*), correlating well with the temporal pattern of telomerase recruitment to the telomere (*Moser et al., 2009*; *Webb and Zakian, 2012*). There-fore, we asked whether the cell cycle-dependent phosphorylation of Ccq1 actually dictates the TEL-patch-mediated shelterin-telomerase interaction and thus restricts telomerase recruitment to late S phase. To test this, we incubated *cdc25-22* cells at non-permissive temperature (36°C) for 3 hr to arrest them in late G2 phase. Cells were then shifted to permissive temperature (25°C) and cell sam-ples were collected every 20 min for co-IP analysis to monitor Tpz1-Trt1 interaction in a 4-hr cell cycle window. As shown in *Figure 6A and C*, the interaction between Tpz1 and Ccq1 remained almost unchanged throughout the cell cycle. Strikingly, Tpz1 and Trt1 association gradually increased after release from the G2 arrest, peaked during late S phase (100–140 min), and then decreased to the level at G2 (*Figure 6A and D*). This temporal pattern of the Tpz1-Trt1 interaction along cell cycle progression correlates very well with that of Ccq1 phosphorylation (*Chang et al., 2013*), and consequently, with that of the Ccq1-Est1 interaction promoted by Ccq1 phosphorylation, which also peaked in late S phase (*Figure 6E and G*). Consistent with our hypothesis, when the criti-cal Rad3[ATR]/Tel1[ATM] phosphorylation site in Ccq1—Thr93 was mutated, neither Tpz1-Trt1 interac-tion (*Figure 6B and D*) nor Ccq1-Est1 interaction (*Figure 6F and G*) peaked in late S phase. Therefore, we propose that through an intermediate telomerase recruitment complex formed by both Ccq1-Est1 and Tpz1-Trt1 interactions (*Figure 7*), the cell cycle information is delivered through Ccq1 Thr93-phosphorylation to the TEL-patch on Tpz1. This in turn enables Tpz1 to directly position telomerase onto the telomere and elongate it during late S phase, after most of the genome has been replicated.

## Discussion

### Telomerase recruitment involving an intermediate state with two-pronged telomere-telomerase interfaces

Our study demonstrates that fission yeast Tpz1 also contains a TEL-patch in its OB-fold domain, analogous to its human ortholog—TPP1 (*Nandakumar et al., 2012*; *Sexton et al., 2012*; *Zhong et al., 2012*). A TEL-patch mutation causes drastically reduced association of telomerase with telomeres, and consequently an *EST* phenotype. Similar phenotypic consequences were previously

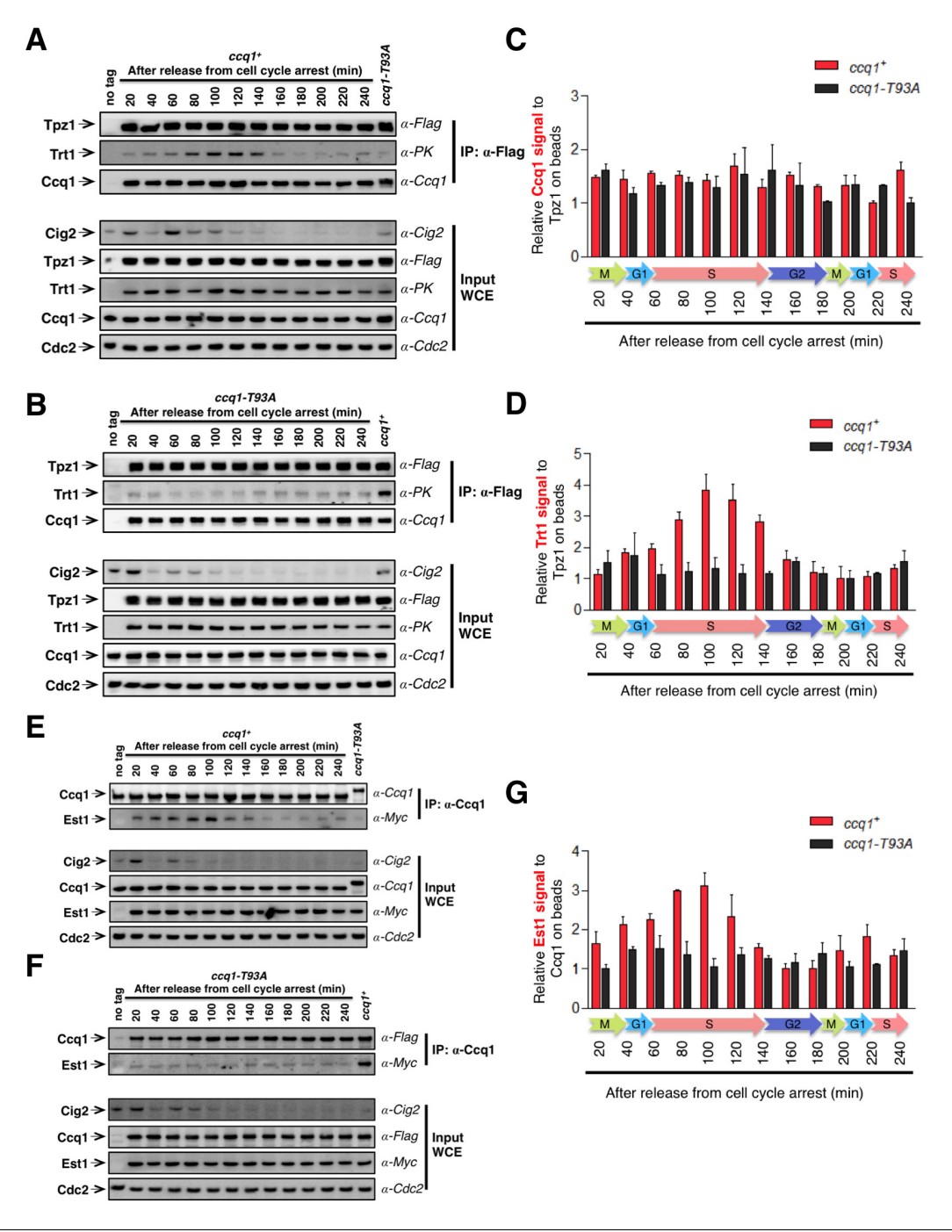

**Figure 6.** Tpz1-Trt1 interaction is cell cycle-regulated. (**A**) and (**B**) Co-IP assays evaluating the binding efficiency of Tpz1-Trt1 and Tpz1-Ccq1 interactions during cell cycle progression in *ccq1⁺* (**A**) and *ccq1-T93A* (**B**) cells. In addition to the indication of time after release from G2 arrest, the levels of S phase cyclin Cig2, which peak at the G1/S boundary and decline to low levels in G2 and M phase, are also shown. Cdc2 was shown as the loading control. Input: 1/30 of input WCE. (**C**) and (**D**) Quantification of the binding efficiency of Tpz1-Trt1 (**C**) and Tpz1-Ccq1 (**D**) interactions during cell cycle progression as assayed in (**A**) and (**B**). The lowest level of Tpz1-Trt1 (**C**) or Tpz1-Ccq1 (**D**) interactions is set to be 1. Plots show mean values ± s.d. for two independent experiments. (**E**) and (**F**) Co-IP assays evaluating the binding efficiency of Ccq1-Est1 interaction during cell cycle progression. As in (**A**) and (**B**), in addition to the indication of time after release from G2 arrest, the levels of S phase cyclin Cig2, are also shown. Cdc2 was shown as the loading control. Input: 1/30 of input WCE. (**G**) Quantification of the binding

*Figure 6 continued on next page*

*Figure 6 continued*

efficiency of Ccq1-Est1 interaction during cell cycle progression as assayed in (E) and (F). The lowest level of Ccq1-Est1 interaction is set to be 1. Plots show mean values ± s.d. for two independent experiments.

reported for the Ccq1 Thr93 phosphorylation-deficient allele (*Moser et al., 2011*; *Yamazaki et al., 2012*). Moreover, our biochemical analyses reveal an unexpected complexity in the Ccq1-Est1 interaction involving two binding sites. As schematically illustrated in *Figure 7*, we envision that in late S phase, when the DNA replication machinery completes most of the genome, Rad3$^{ATR}$/Tel1$^{ATM}$ are activated and phosphorylate the critical Thr93 residue in Ccq1 at telomeres, priming the telomere for telomerase recruitment. Then, telomerase holoenzyme, minimally composed of Trt1-TER1-Est1, is attracted to the telomere by the cell cycle-regulated, phospho-Thr93-mediated Ccq1-Est1 interaction. At the same time, in a collaborative manner, Trt1 interacts with Tpz1 via the respective TEN domain and the TEL-patch residues, thus forming an intermediate telomerase recruitment complex that further engages the telomerase core enzyme (Trt1-TER1) at the very 3' end of the telomere for nucleotide additions. Aided by phospho-Ccq1 Thr93, Est1-Ccq1 interaction at their second binding site takes place, resulting in dissociation of Ccq1 from Tpz1. Because Est1 may use the same surface to interact with Ccq1 as it interacts with TER1 (*Webb and Zakian, 2012*), Est1 is likely to depart from TER1 upon its binding to Ccq1. With phosphorylation of the critical Thr93 in Ccq1 disappearing after late S phase, Ccq1-Est1 interaction diminishes accordingly. Alignment of the very 3' end of telomeric ssDNA to the template region in TER1 might also participate in the telomerase recruitment process, and therefore, only Tpz1 on the extreme 3' end of the telomere, but not the majority of Tpz1 on the internal telomeric regions, is involved in forming the intermediate telomerase recruitment complex.

Ever since the very beginning of the telomere research at the molecular genetics level, the noncatalytic, accessory components of the telomerase holoenzyme, such as Est1p and Est3p in budding yeast, have been demonstrated to play equally important roles in telomere elongation as the catalytic core (telomerase reverse transcriptase and telomerase RNA) (*Lundblad and Szostak, 1989*). Indistinguishable progressive telomere shortening phenotypes are displayed by strains either with deletion of the telomerase RNA, the reverse transcriptase subunit, or accessory proteins (*Leonardi et al., 2008*; *Lingner et al., 1997*; *Lundblad and Szostak, 1989*; *Webb and Zakian, 2008*), suggesting that all these components work collaboratively to form a functional telomerase. Est1, whose homologs have been identified from yeasts to humans, has been proposed to recruit and activate telomerase RNP to telomeres in a cell-cycle dependent manner. Unexpectedly, a previous study discovered that the telomerase core, RNP, Trt1 and TER1 RNA, is also required for the association of Est1 with the telomere (*Webb and Zakian, 2012*). Our finding that telomerase holoenzyme is recruited to telomeres via an intermediate state involving two-pronged cooperative Est1-Ccq1 and Trt1-Tpz1 interactions, uncovers the active role of Trt1 itself in the recruitment process. In addition, our model explains the intricate interdependence between Trt1, TER1, and Est1 for telomeric association of the holoenzyme.

What are the advantages of employing two cooperative telomere-telomerase interactions to recruit telomerase to telomeres? We think that this mode enables temporal and spatial regulation of telomerase recruitment. It has been shown in budding yeast that Tel1$^{ATM}$ prefers shorter telomeres than longer ones. We demonstrate here that the cell cycle-dependent interaction between Ccq1 and Est1 is coupled to the Tpz1 TEL patch-Trt1 interaction. Moreover, in fission yeast, strains with shorter telomeres show a higher level of Ccq1 phosphorylation (*Moser et al., 2011*). Therefore, shorter telomeres could have more access to telomerase due to the higher level of Thr93 phosphorylation in Ccq1. After locating which telomere to elongate, interaction between the TEL-patch of Tpz1 and the catalytic subunit Trt1 further orients the very 3' end of telomeric ssDNA via the sequence-specific Pot1/Tpz1-ssDNA interaction to the active site of Trt1. Two-interface recruitment mode has been observed in other pathways for similar purposes. For example, 53BP1, an important effector of DNA double-strand-break (DSB) response, simultaneously recognizes mononucleosomes containing dimethylated H4K20 (H4K20me2) and H2A ubiquitinated on Lys15 (H2AK15ub) (*Fradet-Turcotte et al., 2013*). There, it was proposed that the engagement of H4K20me2 by the Tudor domain of 53BP1 positions its ubiquitination-dependent recruitment (UDR) motif in the correct orientation to contact

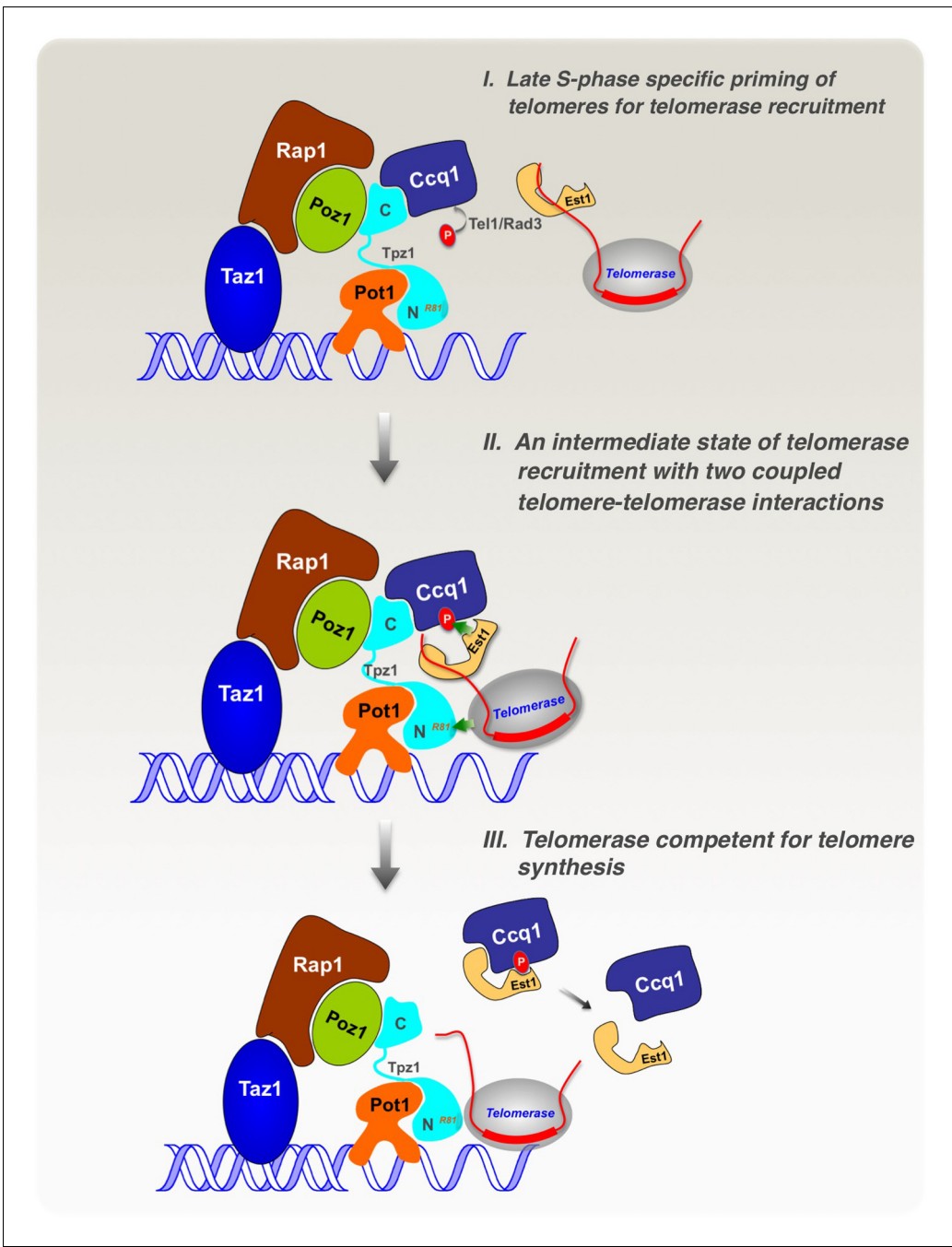

**Figure 7.** Model of cell cycle-regulated telomerase recruitment via an intermediate state with two coupled telomere-telomerase interactions. A model showing telomerase recruitment in fission yeast through an intermediate state in which Trt1 and Est1 in the telomerase holoenzyme are collaboratively anchored to the telomere via Trt1-Tpz1 TEL patch and Est1-Ccq1 interactions, respectively (illustrated by two green arrows drawn in the middle panel), thereby achieving cell cycle-regulated recruitment of telomerase to telomeres.

The following figure supplement is available for figure 7:

**Figure supplement 1.** A speculative two-interface intermediate-state model for human telomerase recruitment.

the epitope formed by H2AK15ub, and thus to ensure that 53BP1 responds only to bona fide DSB signaling.

## Multi-functionality of Tpz1 in regulating telomerase action at telomeres

Within the fission yeast telomere shelterin complex, Tpz1 physically lies in the middle of the telomeric ssDNA and dsDNA binding proteins. Functionally, Tpz1 is positioned between the positive and negative regulators of the telomerase elongation. Tpz1 directly interacts with three other shelterin components: Poz1, Ccq1, and Pot1. This unique position of Tpz1 in the shelterin complex enables its architectural role in shelterin complex assembly and underscores its potential coordination roles in communicating the dsDNA length and/or structural information to the 3′ end of ssDNA—telomerase's ultimate destination. In our previous studies, utilizing biochemically identified Tpz1 separation-of-function mutants that can individually but specifically disrupt Tpz1's interactions with Poz1, Ccq1, or Pot1, we found that Tpz1-mediated complete linkage between telomere dsDNA and ssDNA binding proteins in the shelterin complex is required for defining the telomerase-nonextendible state of telomeres (*Jun et al., 2013*). Disruption of the linkage on either the dsDNA binder or the ssDNA binder side of Tpz1 causes unregulated elongation of telomeres. In addition to maintaining the telomerase-nonextendible state, through its interaction with Ccq1, Tpz1 may also activate the telomerase-nonextendible state of telomeres by participating in breaking down the 'shelterin bridge'. Moreover, Lys75, which is close to the TEL-patch in the OB-fold domain of Tpz1, was demonstrated to have telomerase activation function, the step that follows successful telomerase-telomere association and alters telomerase conformation to become competent for telomere synthesis (*Armstrong et al., 2014*). Recently, SUMOylation of Tpz1-Lys242 in late S phase was shown to enhance the Tpz1-Stn1 interaction, promote Stn1-Ten1 association with telomeres, and thus to coordinate synthesis of the telomeric lagging strand by Polα stimulated via Stn1-Ten1-Polα interaction (*Hannan et al., 2015*; *Miyagawa et al., 2014*). The TEL-patch in Tpz1 and its conserved role in telomerase recruitment characterized in this work further extend the versatility of Tpz1 in telomere length homeostasis. Evidently, the ability to regulate the telomeric state together with the role of recruiting/activating telomerase and coordinating Polα are integrated into one single telomeric protein—Tpz1. This integration ensures timely, accurate, and efficient coupling of conformational transitions of the telomere to the engagement and activation of the telomerase RNP.

## Conservation of telomerase recruitment mechanisms

Although the TEL-patch of the shelterin component TPP1 acting as the telomere-telomerase interface was initially discovered and extensively studied in human cells (*Hockemeyer and Collins, 2015*; *Schmidt and Cech, 2015*), upstream regulatory events of TEL-patch-mediated telomerase recruitment remain to be elucidated. In contrast, regulatory pathways and factors that control telomerase recruitment have been fairly well studied in both budding and fission yeasts due to their convenient and precise genetic manipulability. For instance, in budding yeast, cell cycle-specific assembly and disassembly of active telomerase RNP at telomeres have been shown to restrict telomerase action to late S phase (*Tucey and Lundblad, 2014*). Moreover, telomerase tends to preferentially elongate short telomeres and Tel1$^{ATM}$ is enriched in short telomeres to achieve telomere length homeostasis (*Bianchi and Shore, 2007*; *Sabourin et al., 2007*). Cdk1 was also shown to control the temporal recruitment of telomerase by directing the timing of Cdc13 and Stn1 phosphorylation along cell cycle progression (*Li et al., 2009*; *Liu et al., 2014*). Two recent studies unveil the conserved regulatory role of ATM/ATR in human recruitment and telomere elongation (*Lee et al., 2015*; *Tong et al., 2015*); however, the downstream substrate is mostly unknown.

In this work, we uncovered the conservation of the TEL-patch in fission yeast Tpz1, with similar biochemical and functional roles to human TPP1. This similarity between human and fission yeast makes us speculate that the conservation could probably be extended to other aspects of the telomerase recruitment pathway. Is it possible that human telomerase recruitment also involves two coupled telomere-telomerase interactions? What is the Ccq1-Est1 interaction equivalent in human cells? In fact, human TIN2 (Poz1 homolog) has been demonstrated to be essential for telomerase recruitment (*Abreu et al., 2010*). DC mutations in the C-terminal region of TIN2 (not contained in Poz1) lead to defective association of TIN2 with telomerase (*Frank et al., 2015*; *Yang et al., 2011*). As illustrated in *Figure 7—figure supplement 1*, we speculate that the C-terminal domain of TIN2 is

likely to be the Ccq1-functional equivalent in humans and interacts either with hEST1 or the telomerase RNP directly, forming the second interface between telomeres and telomerase in addition to the TPP1 (TEL-patch)-TERT (TEN domain) interaction. Moreover, TIN2 may also be subject to cell cycle-controlled post-translational modifications, and thus mediate its telomerase recruitment function in a highly regulated way.

## Materials and methods

### Yeast strains, gene tagging, and mutagenesis

Fission yeast strains used in this study are listed in *Supplementary file 1*. Single-mutant strains were constructed by one-step gene replacement of the entire open reading frame (ORF) with the selectable marker. Double- and triple-mutant strains were produced by mating, sporulation, dissection, and selection followed by PCR verification of genotypes. Genes were fused to specific epitope-tags at the C-terminus by homologous recombination; the pFA6a plasmid modules were used as templates for PCR (*Bähler et al., 1998*; *Sato et al., 2005*). Point mutations were made by site-directed mutagenic PCR using the high fidelity polymerase *Pfu* (Agilent). All mutations were confirmed by DNA sequencing (Eton, San Diego, CA). The Trt1-Tpz1 fusion strains were constructed based on previously published strains (*Armstrong et al., 2014*).

### Telomere length analysis

*S. pombe* cells grown in 5 ml YEAU overnight were harvest for genomic DNA extraction. EcoRI-digested genomic DNA was separated on 1% agarose gel at 70 V for 18.5 hr, and then the gels were incubated in 0.25 M hydrochloric acid for 15 min followed by 0.5 M sodium hydroxide and 1.5 M sodium chloride buffer for 30 min and 0.5 M Tris-HCl (pH 7.0) and 1.5 M sodium chloride for 30 min. DNA was transferred to Amersham Hybond-N$^+$ membrane (GE Healthcare Life Sciences) via capillary blotting. DNA was cross-linked to the membrane. The telomeric probe was prepared as previously described (*Jun et al., 2013*). The template of $pol1^+$ was amplified with 5' primer (GGTGCAGAAGACGGTCTG CAAG) and 3' primer (CTTAGCATGCAGAAGCATGCGC), and both probes were labeled by random hexamer labeling using [$\alpha$-$^{32}$P]-dCTP and High Prime (Roche). Hybridizations were carried out with 6 million cpm of probe in Church-Gilbert buffer at 55°C. Blots with both telomeric and $pol1^+$ probe were expose to PhosphorImager screens that were visualized using a Typhoon scanner (GE Healthcare).

### Co-immunoprecipitation

Frozen cell pellets were cryogenically disrupted with FastPrep MP with three pulses (60 s) of bead-beating in ice-cold lysis buffer (50 mM HEPES at pH 7.5, 140 mM NaCl, 15 mM EGTA, 15 mM MgCl$_2$, 0.1% NP40, 0.5 mM Na$_3$VO$_4$, 1 mM NaF, 2 mM PMSF, 2 mM benzamidine, Complete proteinase inhibitor [Roche]). Centrifuge clarified whole cell extracts were adjusted to 13 mg/ml. 300 µl cell extracts were incubated with either anti-Flag M2 affinity gel (Sigma), anti-Ccq1 rabbit serum plus Protein G beads (Roche), or c-myc antibody (Santa Cruz) plus Protein G beads for 2–4 hr at 4°C. The beads were resuspended in SDS loading buffer, boiled, and subjected to western blotting. Western blot analysis was performed using monoclonal anti-Flag (M2-F1804, from Sigma), monoclonal anti-PK (from Abcam), anti-Ccq1 rabbit serum, monoclonal anti-Myc (9E10, from Covance), and anti-Cdc2 (y100.4, from Abcam). 20 µg whole cell extract were used for input control.

### Chromatin immunoprecipitation

Fresh *S. pombe* cells in liquid culture were fixed with 1/10 (vol/vol) ratio of an 11% formaldehyde solution (11% formaldehyde, 100 mM NaCl, 1 mM EDTA at pH 8.0, 0.5 mM EGTA, 50 mM Tris-HCl at pH 8.0) for 20 min, followed by the termination with 125 mM glycine for 5 min. Cell pellets were disrupted in 400 µL of lysis buffer (50 mM Hepes at pH 7.5, 140 mM NaCl, 1 mM EDTA, 1% Trition X-100, 0.1% sodium deoxycholate, Complete proteinase inhibitor [Roche], 1 mM PMSF, 1 mM benzamidine, 1 mM Na$_3$VO$_4$, 1 mM NaF) with FastPrep MP. After three pulse (1 min) of beads-beating, at least 90% cells were broken. Cell extracts were sonicated one time for 30 s in 45 cycles using a Bioruptor (Diagenode). Clarified cell extracts were incubated with anti-Myc resin (9E10, Santa Cruz), anti-Flag M2 affinity gel (Sigma) or anti-Ccq1 rabbit serum followed by protein G-agarose

(Roche, Indianapolis, IN) for 3 hr at 4°C. Then, the beads were washed sequentially, each twice, with lysis buffer, lysis buffer with 500 mM NaCl, wash buffer, and 1x TE buffer. Each sample was added with 100 µl of 10% Chelex100 resin and boiled for 15 min, followed by 20 µg proteinase K treatment for 30 min at 55°C. The recovered DNA were denatured with 0.4 M NaOH and transferred to a Hybond-XL membrane by using a slot module. The blots were hybridized with telomeric probe; the same blot was then re-probed with rDNA probe after stripping off the telomere probe.

## Competitive binding assay

Myc-tagged Est1 under the *nmt1* promoter was expressed in a wild-type strain or a strain expressing Flag-tagged Ccq1, these cells were harvested and cryogenically disrupted to obtain clarified extract followed by co-immunoprecipitation using anti-Flag M2 affinity gel (Sigma) as described above. After incubation, the beads were washed three times with 800 µl lysis buffer. The competition assay was carried out by adding 100 µl lysis buffer with increasing concentrations (5, 100, 2000 nM) of purified recombinant MBP-Tpz1 (406–508) or MBP-Tpz1 (406–508)-L449A to the beads. The mixture was incubated at 4°C for 2 hr with the tube rotating. After extensive washing, the beads were resuspended in SDS-PAGE loading buffer, boiled, and subjected to Western blotting with anti-Flag (M2-F1804, from Sigma), anti-Myc (9E10, from Covance), or anti-MBP (N-17, from Santa Cruz) antibodies.

## Cell cycle synchronization coupled with co-immunoprecipitation

*cdc25-22* cells bearing *ccq1*$^+$ or *ccq1-T93A* grown in 600 ml YEAU overnight at 25°C to OD ~0.3 were shifted to restrictive temperature (36°C) for 3 hr to arrest cells in late G2 phase. Synchronous cultures were then generated by releasing these *cdc25-22* cells to the permissive temperature (25°C). 50 ml cultures were collected every 20 min for 4 hr and were subsequently subjected to co-immunoprecipitation analyses to evaluate Est1-Ccq1 and Tpz1-Trt1 interactions. Anti-Cdc2 (y100.4, from Abcam) was used as a control for input.

## Acknowledgements

We thank Kazunori Tomita, Toru Nakamura, Fuyuki Ishikawa, Julie Cooper, Virginia Zakian, and Takashi Toda for providing plasmids and strains, Kazunori Tomita for advice regarding Trt1 co-IP analysis, Peter Kaiser, Suzanne Sandmeyer and Craig Kaplan for comments on the manuscript and helpful discussions. This work was supported by a Basil O'Connor Starter Scholar Research Award from March of Dimes and an NIH grant R01GM098943 to FQ. HJ is supported by a Predoctoral Fellowship (15PRE22420012) from American Heart Association.

## Additional information

### Funding

| Funder | Grant reference number | Author |
|---|---|---|
| National Institutes of Health | R01GM098943 | Xichan Hu Jinqiang Liu Hyun-IK Jun Feng Qiao |
| American Heart Association | 15PRE22420012 | Hyun-IK Jun |
| March of Dimes Foundation | Basil O'Connor Starter Scholar Research Award | Feng Qiao |

The funders had no role in study design, data collection and interpretation, or the decision to submit the work for publication.

### Author contributions

XH, JL, Acquisition of data, Analysis and interpretation of data, Drafting or revising the article; H-IKJ, J-KK, Acquisition of data, Analysis and interpretation of data; FQ, Conception and design, Analysis and interpretation of data, Drafting or revising the article

## Author ORCIDs

Feng Qiao, http://orcid.org/0000-0002-1704-7257

## Additional files

**Supplementary files**

• Supplementary file 1. Strains used in the study and their genotypes.

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
