## [Decision Letter]

Thank you for submitting your article "Telomerase Recruitment is Promoted by a Bivalent Transition-State Involving both Ccq1 and the TEL-Patch of Tpz1" for consideration by *eLife*. Your article has been favorably evaluated by James Manley (Senior editor) and three reviewers, one of whom, Kathleen Collins, is a member of our Board of Reviewing Editors. The reviewers have discussed the reviews with one another and the Reviewing Editor has drafted this decision to help you prepare a revised submission.

(Note that the Title should be changed as part of the clarifications of essential revision point number 6.)

Summary:

In this thorough and thoughtful study, the authors illuminate interaction trade-offs that unify previously puzzling requirements for telomerase action at telomeres in fission yeast. Using elegant genetic and biochemical approaches, they demonstrate a series of interactions and interaction rearrangements between telomere and telomerase proteins that load and then activate telomerase for end-elongation. Biochemical specificity of interactions accounts for how short telomeres are preferentially elongated in a cell cycle dependent manner. Results from fission yeast studied here provide ideas about telomerase-telomere interaction across species.

Essential revisions:

1) The Tpz1-Ccq1-Est1 ternary complex should be described as speculative in existence or it should be demonstrated directly. Is it possible to directly detect the formation of the ternary complex Est1-Tpz1-Ccq1 using a pull-down of Ccq1 to show that Est1 and Tpz1-C can be pulled down in a Ccq1-dependent manner? Alternatively, since the authors are already using radio-labeled Est1p, perhaps this reagent could be used as the basis for some form of super-shift EMSA experiment to demonstrate formation of the ternary complex?

2) Figure 1 should better document the senescent phenotype that is reported. The authors should analyze the telomere length in the tpz1-R81E/poz1D strain at later generations to make certain that the strain senesces or show that the telomeres are dependent on homologous recombination. In addition, "phenocopying" may not be the correct word to describe similar phenotypes resulting from different mutations.

3) The quantification shown in Figure 2 should normalize Trt1 and Ccq1 in the IP to Tpz1 bound to beads to compensate for mutant effects on protein stability and IP efficiency before making comparisons between strains.

4) Figure 5 experiment should be re-run in a way that normalization with Input levels can be performed, or the statement that Ccq1-T93A and Est1-K252E do not interact should be qualified to reflect the technical limitation. Est1-K252E seems significantly reduced compared to wild type and other Est1 mutants in the ccq1-T93A background.

5) In their concluding model (Figure 7) the authors speculate that Ccq1 and Est1 leave the telomere while Tpz1 stays telomere bound. This data contradicts the study's cell cycle Co-IP data (Figure 6) that shows Tpz1 and Ccq1 are bound together throughout the cell cycle and the Est1-Ccq1 interaction data showing decreasing interaction during late S-phase (Figure 6). In addition, data from the Nakamura lab shows Tpz1- and Ccq1-telomere interaction as co-incident with both protein telomere interaction peaking in late S-phase (Chang et al., 2013), which contradicts this model in which Ccq1 leaves the telomere while Tpz1 remains bound. These discrepancies with the model should be accounted for. Is only a minor fraction of the Tpz1-Ccq1 complex interacting with Est1 at any given time?

6) With respect to the nomenclature used to describe the proposed binding intermediate complex formed between Tpz1-Ccq1-Est1 (telomerase), the reviewers should strongly consider changing the name of the model. The term transition state has a very specific meaning, and is precisely not to be considered a detectable intermediate during a given reaction pathway. In contrast the complex being proposed in the present study, while possibly transient, would be considered a bone fide intermediate complex during telomerase recruitment. In addition, since the paper describes two distinct bivalent interactions, one within the Est1-Ccq1 complex and the second within the putative ternary complex of Tpz1-Ccq1-Est1(telomerase), it is potentially confusing to the reader what exactly is meant by 'bivalent transition state'. The authors might consider revising their wording to something more typical of a transiently formed multi-protein complex (i.e. telomerase pre-priming complex, telomerase recruitment complex, etc.).

[Editors' note: further revisions were requested prior to acceptance, as described below.]

Thank you for resubmitting your work entitled "Telomerase Recruitment is Promoted by an Intermediate-State Involving both Ccq1 and the TEL-Patch of Tpz1" for further consideration at *eLife*. Your revised article has been favorably evaluated by James Manley (Senior editor) and a Reviewing editor.

The manuscript has been improved but there are some remaining issues that need to be addressed before acceptance, as outlined below:

Title:

The title "Telomerase Recruitment is Promoted by an Intermediate-State Involving both Ccq1 and the TEL-Patch of Tpz1" does not sufficiently communicate the import or advance of this work to a general audience. Something like "Multi-step coordination of telomerase recruitment and activation at fission yeast telomeres" extended by something like "through two conformationally coupled interaction interfaces."

Abstract:

Needs to be overhauled to communicate the actual content and importance of this work.

1) "[…]how the TEL-patch communicates with other shelterin components and telomerase holoenzyme to achieve temporally and spatially regulated telomerase recruitment is unclear"

This work is not about how the TEL-patch communicates with other shelterin components or how the TEL-patch communicates to telomerase, which is previously established. This work is about how TEL-patch interaction with telomerase integrates into the overall orchestration of telomerase regulation at chromosome ends.

2) Need to introduce what Ccq1 is, since it is not a household protein name. Could include the adjective telomere-associated or telomere-interacting before first use of Ccq1 in the Abstract.

3) "we discovered that telomerase recruitment is promoted by an intermediate-state.[…]" But recruitment is not "promoted" by an intermediate state – intermediate to what at this point also not explained – instead it is true that 'telomerase action at telomeres requires formation and resolution of an intermediate state.' "[…]to achieve timely and legitimate telomerase engagement" But what is legitimate? No illegitimate engagement is studied here – needs to be something like 'to achieve regulated and efficient telomerase elongation of telomeres.'

Revise "bivalent" and "transition-state"

Definitions of bivalent online: "having two combining sites" "a molecule formed from two or more atoms bound together as a single unitmolecule"

As Reviewers explained previously, the use of "bivalent" is confusing. The revision has not make "bivalent" more clear. It does not make sense to readers even within the telomere field to write "telomerase recruitment is promoted by a bivalent intermediate-state, formed via both Ccq1-Est1 and Tpz1 (TEL-patch)-Trt1 interactions." It would be sufficient and less confusion-inducing to say "is promoted by an intermediate-state with both Ccq1-Est1 and Tpz1 (TEL-patch)-Trt1 interactions."

The Discussion still begins with "A bivalent transition-state model for telomerase recruitment" despite Reviewers' indication to remove "transition-state" and refrain from vague uses of "bivalent."

Hyperbole and coherence

Last paragraph of Intro:

"Thus, the temporal and spatial information for telomerase recruitment is endowed to the TEL-patch through the phosphorylation status of Ccq1 Thr93, thus achieving cell cycle-specific and telomere length-dependent telomere elongation." This work does not direct address "spatial information" or "telomere length-dependent telomere elongation" – here and elsewhere, less vague/out-of-place summary statements would be more useful to understand what this work shows directly. In the Discussion is it appropriate to speculate about how telomere length would affect the two-pronged cooperative Ccq1 and Tpz1 recruitment of telomerase by changing the efficiency of Ccq1 phosphorylation – but there are no experiments that physically dissect length-dependent versus length-independent recruitment in this work.

First paragraph of Results, echoed in last sentence of Results:

"additional interactions between shelterin components and Trt1 must exist to engage Trt1 directly at the very 3' end of the telomere." "this in turn enables Tpz1 to directly position telomerase to the very 3' end of the telomere"

The discussion says that ssDNA might be required to engage Trt1 at the very end of a telomere – that is valid speculation as Discussion section content. But otherwise this work does not address the "very 3' end of the telomere" and therefore does not show a role for shelterin-Trt1 interaction in positioning telomerase at the very 3' end of a telomere. The quoted conclusions, and their like, need to be rephrased or deleted; they lack logic or fulfilment.

Discussion:

Starting from around "Tpz1 may also activate the telomerase-nonextendible state of telomeres through its interaction with Ccq1."

This section needs to be clarified, because return of the telomere to non-extendible state is not described anywhere else and so the ideas need to be spelled out more clearly.

"Moreover, telomerase activation function was also demonstrated for the Tpz1 OB-fold domain (Armstrong et al., 2014), which is close to the TEL-patch."

But the TEL patch is within the OB fold domain. Please spell out the point about activation.

Next paragraph, this sentence is out of place and should be deleted "conservation of the TEL-patch in fission yeast Tpz1 (TPP1 in humans) further supports the similarity of telomerase recruitment pathways in the human and fission yeast systems." The rest of the paragraph is about things other than the TEL patch.

Overall:

This is not an *eLife* paper until it is revised with sentence-by-sentence reflection on the verity of logic and on the accuracy of presentation/typos.

E.g. of just one example of the former, flip first two lines of Introduction for accuracy. "Telomeres are essential[…] In most eukaryotes, telomeres are comprised of short tandem.[…]"

E.g. of just one example of the latter, Dyskeratosis congenital is not a disease.

Also fix the lack of abbreviation definitions (e.g. dsDNA, ssDNA) and noun/verb disagreement or mixed verb tenses.

Would be useful to split the long run-on paragraphs, such as the final paragraph of the Introduction and the first paragraph of the subsection “Est1 and Ccq1 form a stable complex through two binding sites”.

---

## [Author Response]

Essential revisions:

1) The Tpz1-Ccq1-Est1 ternary complex should be described as speculative in existence or it should be demonstrated directly. Is it possible to directly detect the formation of the ternary complex Est1-Tpz1-Ccq1 using a pull-down of Ccq1 to show that Est1 and Tpz1-C can be pulled down in a Ccq1-dependent manner? Alternatively, since the authors are already using radio-labeled Est1p, perhaps this reagent could be used as the basis for some form of super-shift EMSA experiment to demonstrate formation of the ternary complex?

We thank the reviewer for this suggestion and have performed the experiments with results now shown in Figure 5. Neither the active concentration of the previously utilized Est1 (35S-labeled, translated from RRL) nor the in vitro phosphorylation efficiency of Ccq1-T93 by Rad3 appeared to be high enough for the proposed experiment. To overcome these technical limitations, we overexpressed Est1 in *S. pombe* and pulled down Flag-Ccq1—Est1 complex using anti-Flag beads. The complex was then titrated with increasing concentrations of MBP-Tpz1-C (wild-type or Tpz1-L449A mutant that is defective in Tpz1- Ccq1 interaction). At the high concentration of Tpz1, we observed the binding of wild-type Tpz1, but not the Tpz1-L449A mutant, to the Ccq1-Est1 complex. At the same time, decreased association of Est1 with Ccq1 was also observed upon Tpz1 binding. These results indicate that Tpz1-Ccq1-Est1 ternary complex indeed exists but may be less stable than either Ccq1-Est1 or Ccq1-Tpz1 complex, suggesting Tpz1-Ccq1- Est1 be an intermediate state.

*2) Figure 1 should better document the senescent phenotype that is reported. The authors should analyze the telomere length in the tpz1-R81E/poz1D strain at later generations to make certain that the strain senesces or show that the telomeres are dependent on homologous recombination. In addition, "phenocopying" may not be the correct word to describe similar phenotypes resulting from different mutations.*

Following reviewers’ suggestion, we re-streaked the tpz1-R81E/poz1Δ strain for more generations and analyzed its telomere lengths and senescence phenotype at later generations. We found that tpz1-R81E/poz1Δ cells completely lost their telomeres and senesced after generation 150, and survivals containing circularized chromosomes to bypass the need for telomerase-dependent telomere maintenance appeared afterwards. Accordingly, in the revised manuscript, telomere Southern blot in Figure 1 now includes the DNA sample from generation 200, which has circularized chromosome with no telomeric sequence. Evidence of circularized chromosome is shown in Figure 1—figure supplement 3.

3) The quantification shown in Figure 2 should normalize Trt1 and Ccq1 in the IP to Tpz1 bound to beads to compensate for mutant effects on protein stability and IP efficiency before making comparisons between strains.

We did the normalization as suggested and updated Figure 2 accordingly. The result is similar to what was shown in the original submission.

4) Figure 5 experiment should be re-run in a way that normalization with Input levels can be performed, or the statement that Ccq1-T93A and Est1-K252E do not interact should be qualified to reflect the technical limitation. Est1-K252E seems significantly reduced compared to wild type and other Est1 mutants in the ccq1-T93A background.

We agree that the protein level of Est1-K252E in the “Input” is reduced, compared to wild typeand other Est1 mutants in the *ccq1-T93A* background. Therefore, we re-carried out the co-IP experiment with the amount of *est1-K252E/ccq1-T93A* cells three times more than other cells in the assay. As shown in the revised Figure 5, for the *est1-K252E/ccq1-T93A* strain, even with a bit more Est1-K252E protein in the “Input” than other strains assayed in parallel, no Ccq1-Est1 interaction can be detected.

5) In their concluding model (Figure 7) the authors speculate that Ccq1 and Est1 leave the telomere while Tpz1 stays telomere bound. This data contradicts the study's cell cycle Co-IP data (Figure 6) that shows Tpz1 and Ccq1 are bound together throughout the cell cycle and the Est1-Ccq1 interaction data showing decreasing interaction during late S-phase (Figure 6). In addition, data from the Nakamura lab shows Tpz1- and Ccq1-telomere interaction as co-incident with both protein telomere interaction peaking in late S-phase (Chang et al., 2013), which contradicts this model in which Ccq1 leaves the telomere while Tpz1 remains bound. These discrepancies with the model should be accounted for. Is only a minor fraction of the Tpz1-Ccq1 complex interacting with Est1 at any given time?

We thank the reviewers for pointing this out. The speculation that Ccq1 and Est1 leave the telomere is based on the previous yeast 2-hybrid or 3-hybrid results showing that Est1-Ccq1 and Est1-TER1 RNA interactions are mutually exclusive (Armstrong et al., Current Biology 2014) and that Est1 mutants that impair Est1-TER1 RNA interaction also reduce Est1-Ccq1 interaction (Webb and Zakian, Genes & Dev 2012). As the reviewers suggested, since only a very minor fraction of Tpz1-Ccq1 complex, located at the very 3’end of the telomeric ssDNA, interacts productively with Trt1-TER1-Est1 telomerase RNP, the leaving of this small fraction of Ccq1 from telomeres may not be detected by either Ccq1-ChIP or Ccq1- Tpz1 co-IP.

We have now further amended our description and discussion of the model to better emphasize this point, thus avoiding the confusion that readers might have. We also incorporated the cell cycle-dependent Ccq1 phosphorylation and Ccq1-Est1 interaction to the model to illustrate the dissociation of Est1 from Ccq1 when Ccq1 no longer bears phospho-T93.

6) With respect to the nomenclature used to describe the proposed binding intermediate complex formed between Tpz1-Ccq1-Est1 (telomerase), the reviewers should strongly consider changing the name of the model. The term transition state has a very specific meaning, and is precisely not to be considered a detectable intermediate during a given reaction pathway. In contrast the complex being proposed in the present study, while possibly transient, would be considered a bone fide intermediate complex during telomerase recruitment. In addition, since the paper describes two distinct bivalent interactions, one within the Est1-Ccq1 complex and the second within the putative ternary complex of Tpz1-Ccq1-Est1(telomerase), it is potentially confusing to the reader what exactly is meant by 'bivalent transition state'. The authors might consider revising their wording to something more typical of a transiently formed multi-protein complex (i.e. telomerase pre-priming complex, telomerase recruitment complex, etc.).

We agree with the reviewers and have revised the title to “Telomerase Recruitment is Promoted by an Intermediate-State Involving both Ccq1 and the TEL-Patch of Tpz1”.

“Bivalent” is removed from the title and “Transition-state” is changed to “Intermediate-state”. We also provided explanations for “'bivalent” wherever it appears in the manuscript to clarify the two interactions (Ccq1-Est1 and Tpz1-Trt1) involved in recruiting telomerase to telomeres, which is a major new concept in this paper.

[Editors' note: further revisions were requested prior to acceptance, as described below.]

*The manuscript has been improved but there are some remaining issues that need to be addressed before acceptance, as outlined below:*

*Title:*

*The title "Telomerase Recruitment is Promoted by an Intermediate-State Involving both Ccq1 and the TEL-Patch of Tpz1" does not sufficiently communicate the import or advance of this work to a general audience. Something like "Multi-step coordination of telomerase recruitment and activation at fission yeast telomeres" extended by something like "through two conformationally coupled interaction interfaces."*

[…]

*Overall:*

*This is not an eLife paper until it is revised with sentence-by-sentence reflection on the verity of logic and on the accuracy of presentation/typos.*

*E.g. of just one example of the former, flip first two lines of Introduction for accuracy. "Telomeres are essential[…]*

*In most eukaryotes, telomeres are comprised of short tandem.[…]"*

*E.g. of just one example of the latter, Dyskeratosis congenital is not a disease.*

Also fix the lack of abbreviation definitions (e.g. dsDNA, ssDNA) and noun/verb disagreement or mixed verb tenses.

Would be useful to split the long run-on paragraphs, such as the final paragraph of the Introduction and the first paragraph of the subsection “Est1 and Ccq1 form a stable complex through two binding sites”.

We have thoroughly edited the manuscript according to the constructive suggestions and comments provided by the editors/reviewers. We have made changes to incorporate all the points that the editors/reviewers have raised in the text and the figures.